# Parental Ethnicity and Adolescent Development: Evidence from a Nationally Representative Dataset

**DOI:** 10.3390/ijerph20053799

**Published:** 2023-02-21

**Authors:** Lidan Lyu, Danyang Sheng, Yu Chen, Yu Bai

**Affiliations:** 1Center for Population and Development Studies & Center for Family and Gender Studies, Renmin University of China, Beijing 100872, China; 2School of Sociology and Population Studies, Renmin University of China, Beijing 100872, China; 3School of East Asian Studies, University of Sheffield, Sheffield S3 7RA, UK; 4School of Economics & China Institute for Vitalizing Border Areas and Enriching the People (VBEP), Minzu University of China, Beijing 100081, China

**Keywords:** parental ethnicity, academic performance, cognitive competence, health status

## Abstract

Adolescent developmental outcomes can vary significantly by differences in ethnicity. While previous studies have examined the impacts of adolescents’ own ethnicity on their development, little research has been conducted about the impacts of the ethnicity of both parents as an important family background factor which is likely to expose adolescents to a variety of growth environments. Using nationally representative data from the China Family Panel Studies (CFPS) surveys, we examine the relationship between parental ethnicity (including both monoethnic families and interethnic families with intermarried Han and ethnic minority groups) and adolescent developmental outcomes, measured by academic performance, cognitive development, and health. Our results show that adolescents with interethnic parents had higher scores in literacy and mathematics tests than those of monoethnic non-Han parents, but their scores were not statistically significantly different from those in monoethnic Han families. Adolescents with interethnic parents also performed better in fluid intelligence assessments and had lower obesity rates than those with monoethnic ethnic minority parents. Our results further suggest that socioeconomic status, parental education, and education expectations partially mediate the association between interethnic parents and adolescent development. Moreover, parental ethnic composition acts as a potential moderator that influences the effects of parents’ non-agricultural work on adolescent development. Our study expands the growing body of empirical evidence on the relationship between parental ethnicity and adolescent development and is conducive to policy recommendations for interventions in the development of adolescents with ethnic minority parents.

## 1. Introduction

The ethnicity of parents, as an important family attribute, can have a profound influence on the growth and development of adolescents and children. It is widely reported that adolescents from ethnic minority families, as compared to adolescents from other family backgrounds, are disadvantaged in terms of their educational performance, nutritional status, and mental health [1]. In recent years, there has been a rising trend of mixed-ethnic intermarriages. However, it is relatively unknown about the influence of families with interethnic parents on adolescent development. It is likely that interethnic couples might need to resolve differences in values, social norms, and lifestyle choices associated with their respective ethnicities. The collision and fusion of different cultural values and social norms for interethnic parents may influence their children’s development in terms of academic performance, cognitive ability, and even physical fitness [2].

As a multiethnic country with 55 ethnic minorities and the Han majority, China provides a useful case for studying this issue. According to Chinese census data, the proportion of ethnic minorities in the Chinese population has risen from 6.7% in 1982 to 8.9% in 2020, more than doubling on an absolute scale [3]. The proportion of ethnic minorities among children has experienced an even faster growth rate, rising from 7.6% in 1982 to 11.0% in 2020 [4]. From data collected for the seventh census in 2020, the number of interethnic households in China has increased to 14.38 million. Interethnic marriages are increasingly common in China, and more than 26 million people are married outside of their own ethnic groups [5]. A majority of interethnic marriages are between Han and non-Han ethnic minorities, and all 55 ethnic minorities record incidences of interethnic marriages with the Han. For nine ethnic minorities, the number of interethnic marriages with the Han actually exceeds the number of intraethnic marriages within their respective ethnic groups [6,7].

In light of this trend, the goal of this study was to investigate the variations in developmental outcomes of adolescents from families with different ethnic compositions, drawing on nationally representative data from the China Family Panel Studies (CFPS). We used academic performance, cognitive ability, and physical health to measure adolescent development. We further explored the underlying mechanisms driving the observed differences. Specifically, we compared adolescents from monoethnic Han families (Han–Han), monoethnic ethnic minority families (Minority–Minority), and interethnic families involving the Han (Han–Minority). Our research focuses on two specific questions: (1) Does heterogeneity in parental ethnic composition drive differences in adolescent development? (2) How do interethnic marriages, directly and indirectly, affect adolescent development? These are important topics to study because the findings based on China will broaden the scope of research in the literature on the relationship between family background and adolescent development and enrich our understanding of the impact of interethnic marriages on adolescent development. Moreover, our study provides significant insights into interethnic integration and offers useful policy recommendations for the building of inter-embedded multiethnic communities, which are important strategic initiatives for multiethnic countries.

## 2. Related Studies and Hypotheses

Findings from a large number of studies reveal that interethnic marriages can enhance children’s and adolescents’ development in many dimensions. From a biological perspective, children of interethnic Han and non-Han parents may have better physical health due to exogamous unions leading to a very low degree of consanguinity [8]. Related empirical evidence shows that children of interethnic Mongolian–Han couples have better fitness and physical constitution in comparison to children in the same age group but from monoethnic Mongolian or Han families [9]. Interethnic marriages also mitigate the relatively common occurrence of consanguineous marriage found in a number of remotely located ethnic minority groups [10]. However, there are significant cross-sectional differences in educational performance and opportunities for children from different ethnic groups. For example, Chinese children from ethnic minority groups tend to underperform relative to Han children in terms of school attendance, participation in higher education, or academic achievement [6,7]. Strikingly, children from interethnic families tend to outperform their contemporaries from monoethnic families; there is evidence to suggest that children from interethnic families have equal or even better access to learning opportunities [7]. In spite of this, some studies argue that there is no direct relationship between interethnic marriages and children’s academic performance. Instead, the deciding factors that are considered to influence the educational outcome are a child’s personal attributes and the extent of parental support and involvement in their children’s education [9]. In light of the existing literature, this paper proposes Hypothesis 1: there are cross-sectional variations in the adolescent developmental outcomes in terms of academic performance, cognitive ability, and physical health from different parental ethnicity compositions, and adolescents from interethnic families might experience better development due to their growing environments.

Establishing the mechanism of how parental ethnicity affects adolescent development requires numerous considerations. Conjectures and empirical evidence based on social capital theory, cultural capital theory, and human capital theories have long asserted that the socioeconomic background of the family and the extent of parental investment in their children are directly related to adolescent development and access to opportunities [11]. In the literature on this topic, poverty is considered to be a key driver of school dropout rates for ethnic minority children [12]. Children’s education is deemed a consumer product, and poor families cannot afford high education costs, while rich families spend more on their children’s education. Hannum (2002) uses a 1992 Chinese national sample survey to conclude that family poverty is a significant reason for the underperformance of ethnic minority children in school attendance and academic achievements compared to their Han contemporaries [7]. The mechanism of this effect is that ethnic minority children are more likely to drop out of school to earn an income. Furthermore, the level of parental educational attainment positively influences children’s educational performance. Parental investment in education is also positively correlated with children’s educational achievement. Research on the educational investment of ethnic minority children shows that, compared with Han families, ethnic minority families tend to invest less in their children’s education. Using a sample of school dropout rates in children from different ethnic backgrounds in Gansu Province, Sun and Xu (2010) propose that differences in ethnic cultures shape the parents’ perception of whether formal school education is important [13]. They find that parents from ethnic minority groups often view the learning of ethnic culture and receiving school education as mutually incompatible. Their perception is reinforced by local cultural practices specific to an ethnic group, leading to higher dropout rates of ethnic minority children in comparison to Han children. To explore whether such relationships exist, this paper proposes Hypothesis 2: parental educational expectations and education investment have a mediating effect between parents’ ethnic identity and adolescent educational performance.

Parental ethnicity may affect adolescent development through socioeconomic status (SES) and educational expectations. At the same time, variations in parental ethnic composition could also drive differences in the relationship between SES and educational expectation and adolescent growth outcomes. To further understand this heterogeneity and the complex paths of the relationship, this paper incorporates interaction terms to examine the moderating effects of differences in parental ethnic composition on the effect of family SES, parental educational expectations, and investment on children’s health and educational outcomes. The paper proposes Hypothesis 3: the ethnic composition of parents will modify the influence on adolescent development from family socioeconomic status, parental educational expectations, and parental investment.

## 3. Data and Methods

### 3.1. Data

This article is based on data from the China Family Panel Studies (CFPS). The CFPS is a nationally representative survey conducted across 25 provinces or their administrative equivalents, including municipalities and autonomous regions across the country. The CFPS 2010 baseline survey data tracks family members and their gene members identified as of 2010, including their newborn and adopted children, using multistage probability proportional to size (PPS) sampling methods. The population sample in the survey represents approximately 95% of the total population in China and is consistent with the demographic characteristics (such as age and gender) from the 2010 census [14].

As of 2022, there were five waves of published CFPS survey data spanning 2010, 2012, 2014, 2016, and 2018. For adolescent respondents aged from 10 to 15, the CFPS rotates two sets of aptitude tests across waves. The first set tests aptitude in literacy and mathematics and was used in waves 2010, 2014, and 2018. The second set tests memory and number series and was used in waves 2012 and 2016. Basic demographic characteristics, such as respondents’ height, weight, and age, were collected in each wave.

We use pooled cross-sectional data rather than panel data to analyze adolescents’ developmental outcomes, measured by academic performance, cognitive ability, and health. There are two reasons. First, we focus on adolescents aged between 10 and 15 only. The participants in the base year of 2010 were between 14 and 19 years old in the 2014 wave, and most of them were not in our target group in 2014. None of them were included in our study in the 2016 and 2018 waves. Second, as two sets of aptitude tests rotate across waves, there is a four-year gap between the same set of aptitude tests. Therefore, it is difficult to form appropriate panel data for our research purposes. Instead, we combine the five waves of published CFPS data based on information from the child questionnaire, which is used to merge with information on their parents and family collected in the adult, family, and household finance questionnaires, respectively. We only keep the most recent observation for the same adolescent. The steps for processing the data used in our study are summarized below.

Step 1: Data from each wave are processed separately. Each respondent in the child questionnaire is identified by a unique personal code ‘pid’ and also a family code ‘fid’, which allows for the child to be matched with the child’s parents, family, community, and household financial information collected in their respective and separate questionnaires. In the end, we obtain five separate waves of data with adolescents’ personal information, family, and socioeconomic status.

Step 2: The five waves of data from Step 1 are merged using the unique personal code ‘pid’ for each child from the child questionnaire. We match wave-by-wave, beginning with the 2010 wave as the baseline and merging it with the 2012 wave. For those adolescents with more than one observation, we only keep the most recent one. The process is repeated for all subsequent waves until information from all five waves is merged.

Step 3: The merged data are checked for errors and duplicates, and a pooled cross-sectional sample is formed. Given that the study requires data on parental ethnicity, as well as adolescent developmental characteristics, including academic performance, cognitive abilities, and physical health, respondents with missing or inappropriate information for these characteristics are removed from the sample, resulting in a total sample size of 4165 adolescents.

### 3.2. Measures

#### 3.2.1. Parental Ethnicity

Parental ethnicity is determined by ethnicity information reported in the adult questionnaire matched to each adolescent by the child’s unique identifier and family identifier. In our study, we focused on three groups of adolescent households: households with monoethnic Han–Han parents (86.1%), households with interethnic Han–Minority parents (4.6%), and households with monoethnic Minority–Minority parents (9.0%), as shown in Table 1. Interethnic marriages involving non-Han minorities also exist. However, the sample size of 16 observations of adolescents from such interethnic minority households is so small that we do not include them in the analysis.

#### 3.2.2. Adolescent Development

Adolescents’ overall developmental status is measured by three factors: academic performance, cognitive development, and physical health. The CFPS has two sets of adolescent aptitude tests (literacy and mathematics vs. memory and number series) for respondents based on questions from the US Health and Retirement Survey (HRS). Existing research suggests that scores for literacy and mathematics tests capture crystallized intelligence, while scores on memory and number series tests capture fluid intelligence [15]. Therefore, we interpret CFPS scores on the literacy/word test (0–34 points) and mathematical test (0–24 points) to reflect adolescents’ crystallized intelligence and scores on the memory test (0–10 points) and number series test (0–15 points) to reflect fluid intelligence. Since crystallized intelligence tends to increase with age and years of schooling, we control for adolescents’ years of schooling in related analyses. On the other hand, fluid intelligence remains relatively stable over time. Physical health is measured by BMI z-scores following the WHO’s definition, in which adolescents with BMI below the 15th percentile are defined as underweight, those between the 15th and 85th percentiles as normal, those between the 85th and 95th percentiles as overweight, and those at or above the 95th percentile as obese. The obesity rate for children in China has been increasing rapidly since 2010, making childhood obesity a serious concern [16]. As a result, our analysis treats the obese and overweight categories as one group and treats the normal and underweight categories as the control group to shed light on the factors influencing overweight and obesity in adolescents.

#### 3.2.3. Family Socioeconomic Status and Child Investment

Family socioeconomic status is a function of family income, parent’s employment status, parental education attainment, and hukou type. Family income is measured by the annual comparable gross household income from the CFPS data. Parental employment status is determined by whether the parents are working and whether they are engaged in agricultural work. In the CFPS adult questionnaire, question QG3 asks, “Do you currently have a job?” If the answer is affirmative, the corresponding response is recorded as “Yes”; if otherwise, it is recorded as “No”. The questionnaire further provides a binary response variable that records whether individuals are engaged in agricultural vs. non-agricultural work. The level of parental education attainment or years of schooling is measured by the maximum of either parent’s years of schooling. In China, a household’s hukou type is closely linked to a family’s socioeconomic status. According to the hukou (household registration) system, Chinese citizens have been registered with either agricultural (rural) or non-agricultural (urban) hukou status at a particular place since they were born. People with different hukou statuses are entitled to different social benefits and services. We, therefore, include the father’s and mother’s hukou status in our analysis of family socioeconomic characteristics. To assess parental investment, we use parental educational expectations and household spending on education as measures. Previous studies have used the attendance at junior college as the threshold to define whether parents have high or low expectations on their children’s educational outcomes [17]. We follow the literature in measuring parental expectations by using the CFPS response variable on whether the parents expect their children to obtain at least a junior college diploma. The CFPS also asks parents about the household’s total expenditure on their children’s education over the previous 12 months and whether their children attended after-school tutoring. The proportion of family income spent on education reflects the importance that parents place on their children’s education. We, therefore, use their responses to educational expenditure and after-school tutoring attendance to measure parental investment in their children’s education.

#### 3.2.4. Population Demographics

We also construct basic demographic information on the adolescents, such as gender (male or female), age, family size, number of siblings, and regional characteristics (town or country), and labeled the year of the wave corresponding to when each observation is collected.

### 3.3. Statistical Analysis

Stata 16.0 is used for the data analysis. We analyze the relationship between parental ethnicity with adolescent development along three dimensions. First, we investigate whether the parental ethnic composition is directly related to adolescent development to explain variations in educational performance, cognitive development, and physical health among adolescents in order to address Hypothesis 1. Second, we examine whether family socioeconomic status and parents’ educational expectations and investment are potential mediating mechanisms that explain the relationship between parental ethnic composition and adolescent development in order to address Hypothesis 2. Third, regarding Hypothesis 3, we explore the role of parental ethnic composition as a potential moderator variable that affects the association between adolescent development and family socioeconomic status, parents’ educational expectations, and parental investment (see Figure 1).

The empirical analysis in this paper is therefore divided into three steps based on the theoretical framework and data characteristics:(1)Use the multiple linear regression model and the multinomial logit model to test the impact of parental ethnic composition on adolescent educational/cognitive development and health, respectively. With this methodology, we can control for factors such as family socioeconomic status and educational expectations in order to examine the robustness of the impact of ethnic composition on adolescent development. Before performing the regression analysis, the multicollinearity diagnosis is carried out on the explanatory variables. The variance inflation factor (VIF) of each variable is less than 10, indicating that there is no serious multicollinearity problem.(2)Use the mediation model to test whether family socioeconomic status, parents’ educational expectations and investment, and children’s health status are intervening variables that transmit mediating effects of parents’ ethnic composition on adolescents’ educational performance. We interpret model parameter estimates to evaluate the statistical significance and direction of the path coefficients.(3)Create interaction terms between parental ethnic composition with other explanatory variables to evaluate whether parental ethnic composition acts as a potential moderator that changes the effects of other explanatory variables on adolescent development.

## 4. Results

### 4.1. Parental Ethnicity and Development of Adolescents: Comparisons between Adolescents of Different Parental Ethnicity

Table 2 shows the descriptive statistics of adolescent attributes across different parental ethnic compositions. Among adolescents from monoethnic Han–Han families (Han–Han adolescents), interethnic Han–Minority families (Han–Minority adolescents), and monoethnic Minority–Minority families (Minority–Minority adolescents), we observe heterogeneity in developmental measures. Education-related outcomes are reported in terms of the various aptitude test scores. Han–Han adolescents perform the best in literacy (Wordtest score), while Han–Minority adolescents rank highest in mathematics (Mathtest score); both groups are very close in both tests, with differences of only 0.18 points and 0.13 points in the Wordtest and Mathtest scores, respectively. Both scores for Minority–Minority adolescents are significantly lower. Their average Wordtest and Mathtest scores are 4.42 points and 1.76 points less than the Han–Han group and 4.24 points and 1.89 points less than the Han–Minority group. The Han–Minority adolescents also perform relatively well in fluid intelligence as measured by their memory and number series test scores, which differ from the Han–Han adolescents by only 0.24 points and 0.26 points but are 0.48 points and 1.85 points higher than Minority–Minority adolescents. Minority–Minority adolescents also differ from the other two groups in physical health, proxied by their BMI z-scores. The proportion of obese adolescents in Minority–Minority families is 5.52% higher than those in Han–Han families, while the proportion of obesity in Han–Minority adolescents is 4.41% higher than Han–Han adolescents.

There is also a disparity in family socioeconomic attributes. The average family income of Han–Minority families is 41,257 yuan, which is 5400 yuan less than Han–Han families but 10,311 yuan higher than Minority–Minority families. Han–Minority parents have the highest average years of education, with an average of 8.83 years, which is 0.11 years more than Han–Han parents and 2.86 years more than Minority–Minority parents. Han–Minority parents are similar to Han–Han parents in terms of the proportion of employment in non-agricultural jobs by either or both of the parents, which are 65.75% and 65.66%, respectively; for the Minority–Minority group, the proportion is the lowest at 43.01%. Over a quarter of Han–Minority parents hold non-rural hukou by either or both parents, the highest proportion among the three groups. The proportion of Han–Han parents is slightly lower at 24.92%, while Minority–Minority parents hold the least at less than 10%.

All three groups have similar expectations of their children’s educational attainment. Han–Minority parents have the highest expectations, with 79.56% expecting their children to obtain at least a junior college diploma. 76.41% of Han–Han parents report similar expectations and only 71.77% of Minority–Minority parents do so. Consistent with the higher expectations, Han–Minority parents also invest more in their children’s education, spending 10.76% of the family income on related expenditures in the past 12 months; in comparison, related expenditures by Han–Han and Minority–Minority families are 8.39% and 7.14%, respectively. More of the Han–Minority adolescents attend after-school tutoring (24.86%) compared to the other two groups (18.30% of Han–Han adolescents and 5.38% of Minority–Minority adolescents); attendance of the Minority–Minority group is notably low and less than a quarter of the Han–Minority group.

### 4.2. Regression Results

#### 4.2.1. Parental Ethnicity and Adolescent Academic Performance

Columns 2 and 3 of Table 3 report the relationship between parental ethnicity and adolescent academic achievement. After controlling for family socioeconomic factors, educational expectations and expenditure, and adolescents’ individual characteristics, parental ethnicity is significantly and positively related to adolescent academic performance. The parental ethnic composition is associated with a *p*-value of less than 0.05 in explaining both literacy (Wordtest) and mathematics (Mathtest) scores. Han–Minority adolescents outperformed Minority–Minority adolescents on the literacy test by 2.2 points (*p* < 0.001) and the math test by 1.03 points (*p* < 0.05). Han–Han adolescents also scored significantly better on both tests than the Minority–Minority group. If the reference group is switched to the Han–Han adolescents, the differences between the Han–Han and Han–Minority groups become statistically insignificant, whereas the Minority–Minority adolescents scored significantly lower in both literacy and math scores.

Parental education attainment, non-agricultural employment, parental educational expectations for their children, and after-school tutoring attendance are all significantly and positively correlated with literacy and math test scores. This result indicates that higher socioeconomic status and higher parental investment, as reflected by educational expectations and education investment, have a positive and significant impact on adolescent academic performance.

Columns 4 and 5 of Table 3 show the relationship between parental ethnicity and adolescent cognitive development. In contrast to academic performance, there is no statistically significant difference in memory test scores between Han–Minority adolescents and Minority–Minority adolescents. Han–Han adolescents score 0.524 points higher in the memory test compared to Minority–Minority adolescents (*p* < 0.001). On the other hand, Han–Minority adolescents score 1.196 points higher in the number series test than the Minority–Minority group (*p* < 0.001). Han–Han adolescents also score higher than the Minority–Minority group. Most family socioeconomic factors and parental investment do not show a significantly strong relationship with memory test scores (*p* < 0.1), thus confirming the view that fluid intelligence is a cognitive ability that is a function of neural development. In contrast, family income, non-agricultural employment, non-rural hukou, educational expectations, and educational investment are all significantly and positively related to number series test scores, which also improve with age and years of education. Parental ethnicity clearly has a significant impact on adolescent fluid intelligence.

Column 6 of Table 3 presents the relationship between parental ethnicity and adolescent obesity. The incidence of obesity does not appear to be significantly different between Han–Minority adolescents compared to Minority–Minority adolescents. However, Han–Han adolescents exhibit a lower incidence at only 55% of that of Minority–Minority adolescents (*p* < 0.001). When the reference group is switched to the Han–Han adolescents, there is no significant difference in the incidence of obesity between Han–Han vs. Han–Minority adolescents. Family socioeconomic attributes do not appear to be related to the incidence of adolescent obesity.

#### 4.2.2. Mediating Role of Household Characteristic between Parental Ethnicity and Adolescent Development

We note that there exist significant differences in the household characteristics across the three groups of parental ethnicities, and family socioeconomic factors have a statistically significant relationship with adolescent developmental outcomes. As a result, we used the causal-steps approach to test whether family socioeconomic factors act as mediators between parental ethnic composition and adolescent development. We used bootstrapping to test the significance of the indirect effect to relax the Sobel test’s inherent assumption of normal distribution.

Table 4 presents the results of the mediation analysis. Given that socioeconomic factors, as well as educational expectation and investment, are not significantly related to the memory test score and the incidence of obesity (*p* > 0.05), we only show the mediating analysis for academic performance and number series test score. We find that the bias-corrected bootstrap confidence for the product of path coefficients of family income and share of family income spent on education contains zero. In contrast, those for the remaining family socioeconomic factors, as well as educational expectation and investment, do not contain zero, thus indicating that the mediation effect from these factors is significantly different from zero. As expected, parental ethnicity affects adolescent academic performance (literacy and math test scores) through family socioeconomic factors, as well as educational expectations and investment. In terms of number series test scores, which are a measure of fluid intelligence that is relatively stable over time and age, the effect of parental ethnicity on Han–Minority and Han–Han adolescents’ scores also operates via mediating pathways through the extent of parental education attainment and level of family income. For Han–Han adolescents, the mediating effect of whether parents have non-agricultural employment is not significant.

#### 4.2.3. Moderating Role of Parental Ethnicity on the Development of Adolescent

Table 5 shows the results of the moderation test of parental ethnicity on the relationship between family socioeconomic status and adolescents’ academic performance. We add multiple interaction terms between parental ethnic composition and parents’ years of schooling, family income, non-agricultural work, and non-agricultural hukou to analyze the extent to which parental ethnic composition has a moderating effect on literacy and mathematics test scores. The results show that, for the literacy test (Wordtest), statistically significant coefficients are found on the interaction terms between Han–Han adolescents and parents’ years of schooling, non-agricultural work, and educational expectations, thus suggesting the existence of a significant moderation effect. Compared to adolescents from Minority–Minority families, the more years of schooling obtained by the parents of Han–Han families, the higher these Han–Han adolescents score on the word test. In contrast, being adolescents from Han–Han families weakens the positive influence that non-agricultural work and higher education expectation have on word test scores. This negative effect is also consistent in the set of results from the mathematics test scores (Mathtest), in which the coefficients of interaction terms for Han–Han parents vs. non-rural hukou, and Han–Han vs. education expectation are all significantly negatively, suggesting that having Han–Han parents reduces the positive effect of non-agricultural work and non-rural hukou on adolescents’ performance on the mathematics test. The findings are consistent with earlier work by Hong Yanbi (2010).

Table 6 reports the results of the moderation test of parental ethnicity on the relationship between family socioeconomic status and adolescents’ cognitive development. Based on the analysis on both the memory and number series tests, the interaction term between parental ethnicity and non-agricultural work is the only term to show statistical significance. For the memory test component, the coefficient on the Han–Han parents interacted with non-agricultural work is negative, suggesting that compared to the Minority–Minority group from families with at least one parent being employed in non-agricultural jobs, the general positive effect of non-agricultural work on adolescent memory ability is weakened for those with Han–Han parents. For the number series test, the interaction term of Han–Minority parents with non-agricultural work has a statistically significant and positive coefficient, indicating that parental ethnicity positively moderates the effect of non-agricultural work on adolescents’ numerical logic. Specifically, compared to the Minority–Minority group, adolescents from Han–Minority families for which at least one parent holds non-agricultural work have better numerical logic results. Since we do not find statistically significant relationships connecting family socioeconomic factors with adolescent obesity rates, a moderation test along this dimension is not conducted.

## 5. Conclusions

Using data from China’s nationally representative CFPS survey across five waves, this study conducted a set of empirical analyses on the influence of parental ethnicity composition on adolescent development, as well as the underlying mechanism. By exploring the differences in the developmental outcomes of adolescents from monoethnic families with Han–Han parents, interethnic families with Han–Minority parents, and monoethnic families with Minority–Minority parents, this study established that, after controlling for differences in family socioeconomic factors and individual characteristics, there are significant associations between parental ethnicity composition and adolescent academic performance, cognitive development, and health. Adolescents from families with interethnic Han–Minority parents perform better in literacy, mathematics, and number series tests in comparison to adolescents from families with interethnic Minority–Minority parents. Adolescents from interethnic Han–Minority families have similar educational outcomes as those from monoethnic Han–Han families.

This study contributes to existing knowledge and practice in three aspects. First, the analysis of the impacts of the ethnicity of both parents as an important family characteristic on adolescent educational performance provides depth and breadth to existing research that focuses only on the ethnicity of adolescents. Adopting the perspective of the family and parental ethnicity beyond the adolescent’s individual characteristics allows for a more robust investigation into how a range of attributes associated with different ethnic groups can influence adolescent development. Second, the study explores the pathways between parental ethnicity and adolescent development for a detailed analysis of the direct and indirect effects. We find that parental ethnicity has moderating effects on some family socioeconomic factors, as well as parental education expectations and investment. We further show that the heterogeneity in developmental outcomes of adolescents from families with different parental ethnicity compositions could be partly explained by indirect mediating influences from differences in family socioeconomic status, parental education expectations, and parental investment in children’s education. Third, our findings can be applied in practice with important policy implications. Interethnic marriages are proliferating in numbers. By understanding the factors that influence the educational performance of adolescents from families with different ethnic compositions and delineating the pathways of such influences, we can better appreciate the effects of interethnic marriages on adolescent development, as we find adolescents from interethnic Han–Minority families have better academic performance and cognitive development. On the other hand, our results indicate that more efforts are required to support adolescents from monoethnic Minority–Minority families, whose relative family socioeconomic status, as well as parental expectation and investment, is weaker compared to that of the other two groups. Notably, either or both parents being employed in non-agricultural jobs has a positive impact on adolescent educational performance. Policy initiatives can target family socioeconomic status and parental investment to support and enhance the developmental outcomes of adolescents from ethnic minority families. Efforts should be made to accelerate the economic and social development in ethnic minority areas and to improve the quality of life for minority families. It would be useful to develop policy initiatives aimed at improving the non-agricultural employment rate of ethnic minorities, which would improve their household income. Moreover, the government should promote the education programs for ethnic minorities and pay more attention to the cultivation of talents among ethnic minorities.

Our study advances the existing understanding of the relationship between parental ethnicity and adolescent development, but it has limitations. Firstly, our sample population is restricted to adolescents aged 10 to 15 who are the target group in the CFPS survey. As a result of the relatively narrow age range, the sample cannot represent adolescents of all age groups in China. Secondly, this study treats ethnic minorities as an aggregate group, given the limited sample size. The conclusions drawn, therefore, reflect the average effects on the general ethnic minority population, while we keep in mind that there exist differences in religious beliefs, cultural values, and socioeconomic status among different ethnic minority groups in China. Future research could explore the impacts of such differences among different ethnic minorities on adolescent development. In addition, there are large disparities in the population size across different ethnic minority groups. The sample of ethnic minorities in this study may not be nationally representative of all ethnic minorities. Finally, while two different assessment outcomes are used to measure academic performance and cognitive development to ensure a more comprehensive measurement of adolescent developmental outcomes, health is measured only through adolescent obesity. Future research can provide a more systematic and detailed analysis of this dimension by incorporating both physical and mental health attributes.

## Figures and Tables

**Figure 1 ijerph-20-03799-f001:**
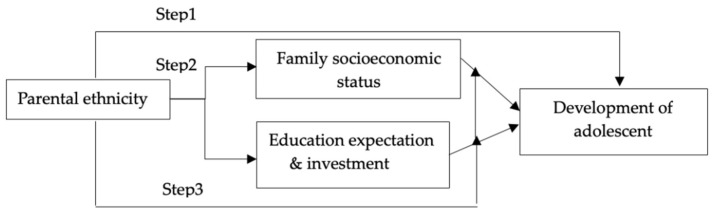
Three steps of statistical analysis.

**Table 1 ijerph-20-03799-t001:** Sample distribution.

	Num.	Percent
Parental ethnicity of adolescents		
Han parents	3416	86.1%
Interethnic parents	181	4.6%
Ethnic parents	356	9.0%
Parents have different Ethnic minority	16	0.4%
Total	4165	100%
Year		
2010	910	21.9%
2012	704	16.9%
2014	914	21.9%
2016	995	23.9%
2018	642	15.4%
Total	4165	100%

**Table 2 ijerph-20-03799-t002:** Individual adolescent and household characteristics.

Individual and Household Characteristics	Han–Han Parents	Han–Minority Parents	Minority–Minority Parents
Wordtest score	21.38	21.20	16.96
Mathtest score	10.22	10.35	8.46
Memory test score	5.57	5.33	4.85
Number series test score	9.27	9.01	7.16
BMI_z, 1 = obesity	14.37%	18.78%	19.89%
Family income(yuan)	45,757.1	41,257.1	30,946.0
Parents’ years of schooling	8.72	8.83	5.97
Either or both parents with non-agricultural work, 1 = yes	65.66%	65.75%	43.01%
Either or both parents with non-rural hukou, 1 = yes	24.92%	26.63%	9.06%
Education expectation (>junior college), 1 = yes	76.41%	79.56%	71.77 %
Share of family income spent on education	8.39	10.76	7.14
Proportion of after-school tutoring attendance	18.30	24.86	5.38
Gender, 1 = female	46.31	18.30	48.12
Age	11.89	11.80	11.94
Family size	4.91	4.51	4.51
Number of siblings	1.00	0.82	1.58
Type of residence (1 = urban)	38.06	40.33	20.70

**Table 3 ijerph-20-03799-t003:** Parental ethnicity and adolescent development.

	Academic Performance (OLS)		Cognitive Development (OLS)		Health (Logistic)
Variable	Wordtest	Mathtest	Memory Test	Number Series Test	Incidence of Obesity
Han–Minority	2.209 ***	1.029 **	0.269	1.196 **	0.582
	(2.988)	(2.474)	(0.278)	(0.493)	(−1.62)
Han–Han	2.524 ***	0.734 ***	0.524 ***	1.146 ***	0.550 **
	(5.663)	(2.930)	(0.177)	(0.315)	(−3.08)
Parents’ years of schooling	0.194 ***	0.100 ***	0.0117	0.0542 *	0.992
	(4.605)	(4.213)	(0.0159)	(0.0282)	(−0.44)
Family income (log)	0.205	0.153 *	0.0141	0.492 ***	0.918
	(1.269)	(1.677)	(0.0550)	(0.0978)	(−1.31)
Either or both parents with non-agricultural work (yes = 1)	0.763 **	0.753 ***	−0.0161	0.449 **	0.975
	(2.474)	(4.342)	(0.106)	(0.188)	(−0.19)
Either or both parents with non-rural hukou (yes = 1)	0.370	0.232	0.238 *	0.525 **	1.154
	(1.046)	(1.165)	(0.132)	(0.235)	−0.91
Education expectation (>junior college = 1)	1.612 ***	1.075 ***	0.213 *	1.033 ***	1.012
	(4.182)	(4.960)	(0.115)	(0.204)	−0.09
Share of family income spent on education	0.0123	0.0055	0.007 *	0.0197 ***	1.001
	(1.133)	(0.911)	(0.00388)	(0.00690)	−0.19
Proportion of after-school tutoring attendance (yes = 1)	1.490 ***	0.467 **	0.0302	0.164	-
	(4.029)	(2.244)	(0.137)	(0.244)	-
Age (at the time of survey)	1.841 ***	1.345 ***	−0.008	0.222 ***	0.624 ***
	2.209 ***	1.029 **	(0.0377)	(0.0669)	(−9.11)
School grade	1.868 ***	0.983 ***	0.110 **	0.196 **	1.063
	(−9.279)	(−8.299)	(0.0472)	(0.0838)	−1.56
Gender (female = 1)	1.134 ***	0.178	0.102	−0.393 **	0.375 ***
	(4.402)	(1.227)	(0.0954)	(0.169)	(−7.85)
Family size	−0.249 ***	−0.144 ***	0.0295	−0.0644	1.029
	(−2.662)	(−2.733)	(0.0322)	(0.0573)	−0.72
Number of siblings	−0.842 ***	−0.271 ***	−0.165 **	−0.188	0.915
	(−4.553)	(−2.605)	(0.0670)	(0.119)	(−1.00)
Wave of survey	0.265 ***	−0.096 *	0.244 *	0.119	1.214
	(2.914)	(−1.869)	(0.127)	(0.225)	−1.75
Constant	−8.350 ***	−9.317 ***	3.570 ***	−1.659	110.285 ***
	(−4.534)	(−8.996)	(0.678)	(1.205)	(5.62)
Observations	2281	2281	1724	1724	2489
R-squared (Pseudo R2)	0.283	0.347	0.036	0.133	0.939
adj_R2	0.343	0.343	0.027	0.125	

Note: *** *p* < 0.01, ** *p* < 0.05, and * *p* < 0.1.

**Table 4 ijerph-20-03799-t004:** Mediation analysis and 95% bias-corrected confidence interval.

Pathway	*B*	SE	95%CI	
**1. Mediating effect on academic performance—Wordtest score**				
Han–Minority→parents’ years of schooling→Wordtest score	0.65 ***	0.16	0.34	0.96
Han–Han→parents’ years of schooling→Wordtest score	0.39 ***	0.09	0.21	0.57
Han–Minority→family income→Wordtest score	0.13 *	0.74	−0.02	0.28
Han–Han→family income→Wordtest score	0.20 ***	0.06	0.08	0.32
Han–Minority→non-agricultural work→Wordtest score	0.29 ***	0.94	0.10	0.47
Han–Han→non-agricultural work→Wordtest score	0.17 ***	0.05	0.02	0.36
Han–Minority→education expectation→Wordtest score	0.26**	0.11	0.05	0.47
Han–Han→education expectation→Wordtest score	0.13 **	0.60	0.01	0.25
Han–Han→share of family income spent on education→Wordtest score (insignificant)	0.02	0.04	−0.06	0.11
Han–Minority→share of family income spent on education→Wordtest score (insignificant)	−0.01	0.02	−0.04	0.02
Han–Minority→after-school tutoring→Wordtest score	0.30 ***	0.11	0.09	0.51
Han–Han→after-school tutoring→Wordtest score	0.05	0.04	−0.04	0.13
**2. Mediating effect on academic performance—Mathtest score**				
Han–Minority→parents’ years of schooling→Mathtest score	0.38 ***	0.10	0.19	0.53
Han–Han→parents’ years of schooling→Mathtest score	0.22 ***	0.05	0.12	0.33
Han–Minority→family income→Mathtest score	0.09	0.51	−0.01	0.19
Han–Han→family income→Mathtest score	0.14 ***	0.04	0.07	0.21
Han–Minority→non-agricultural work→Mathtest score	0.22 ***	0.07	0.09	0.36
Han–Han→non-agricultural work→Mathtest score	0.13 ***	0.04	0.06	0.20
Han–Minority→education expectation→Mathtest score	0.16 **	0.07	0.02	0.30
Han–Han→education expectation→Mathtest score	0.08 **	0.04	0.01	0.16
Han–Han→share of family income spent on education→Mathtest score (insignificant)	−0.01	0.03	−0.05	0.05
Han–Minority→share of family income spent on education→Mathtest score (insignificant)	0.00	0.01	−0.02	0.02
Han–Minority→after-school tutoring→Mathtest score	0.15 ***	0.05	0.04	0.25
Han–Han→after-school tutoring→Mathtest score	0.02	0.02	−0.02	0.07
**3. Mediating effect on cognitive development—Number series test score**				
Han–Minority→highest year of parental education attainment→number series test score	0.33 ***	0.09	0.16	0.50
Han–Han→highest year of parental education attainment→number series test score	0.28 ***	0.06	0.16	0.40
Han–Minority→family income→number series test score	0.88 **	0.03	0.02	0.16
Han–Han→family income→number series test score	0.09 **	0.03	0.02	0.16
Han–Minority→non-agricultural work→number series test score	0.90	0.06	−0.04	0.22
Han–Han→non-agricultural work→number series test score	0.11 ***	0.04	0.03	0.18
Han–Minority→education expectation→number series test score	0.07	0.05	−0.03	0.16
Han–Han→education expectation→number series test score	0.01	0.02	−0.05	0.05
Han–Han→share of family income spent on education→number series test score (insignificant)	0.01	0.01	−0.02	0.03
Han–Minority→share of family income spent on education→number series test score (insignificant)	0.01	0.01	−0.01	0.01
Han–Minority→after-school tutoring→number series test score	0.09 *	0.05	−0.01	0.18
Han–Han→after-school tutoring→number series test score	0.02	0.02	−0.02	0.05

Note: *** *p* < 0.01, ** *p* < 0.05, and * *p* < 0.1.

**Table 5 ijerph-20-03799-t005:** Moderator effect analysis: literacy and mathematics tests.

Variables	Wordtest				Mathtest			
	(1)	(2)	(3)	(4)	(1)	(2)	(3)	(4)
Parental ethnicity **(Minority–Minority = 0)**								
Han–Minority	2.971 *	2.924 **	1.714 **	2.860	1.062	1.778 ***	1.126 **	4.058 ***
	(1.734)	(2.390)	(1.979)	(1.292)	(1.102)	(2.584)	(2.316)	(3.275)
Han–Han	1.152	3.261 ***	2.569 ***	4.047 ***	0.158	0.686 **	0.918 ***	1.008 *
	(1.435)	(5.542)	(5.460)	(4.282)	−0.35	(2.075)	(3.474)	(1.905)
Han–Minority × Parents’ years of schooling	−0.031				0.019			
	(−0.169)				(0.183)			
Han–Han × Parents’ years of schooling	0.198 **				0.083			
	(1.971)				(1.466)			
Han–Minority × Non-agricultural work		−1.653				−1.070		
		(−1.069)				(−1.231)		
Han–Minority × Non-agricultural work		−1.743 **				0.041		
		(−1.993)				(0.083)		
Han–Minority × Non-rural hukou			1.108				−1.562	
			(0.593)				(−1.487)	
Han–Han × Non-rural hukou			−0.545				−1.890 **	
			(−0.381)				(−2.352)	
Han–Minority × Education expectation				−0.940				−3.378 **
				(−0.402)				(−2.578)
Han–Han × Education expectation				−1.950 *				−0.383
				(−1.847)				(−0.648)
Constant	−6.904 ***	−8.432 ***	−7.864 ***	−8.364 ***	−8.578 ***	−8.954 ***	−9.100 ***	−9.182 ***
	(−3.605)	(−4.485)	(−4.237)	(−4.292)	(−7.962)	(−8.470)	(−8.729)	(−8.396)
Control	Yes	Yes	Yes	Yes	Yes	Yes	Yes	Yes
Observation	2281	2281	2281	2281	2281	2281	2281	2281
R-squared	0.286	0.285	0.285	0.280	0.350	0.350	0.351	0.351

Notes: t-statistics in parentheses, *** *p* < 0.01, ** *p* < 0.05, and * *p* < 0.1. Coefficients on the main effects are estimated but not reported in the paper to focus on the interaction terms.

**Table 6 ijerph-20-03799-t006:** Moderator effect analysis: memory and number series tests.

Variables	Memory Test				Number Series Test			
	(1)	(2)	(3)	(4)	(1)	(2)	(3)	(4)
Han–Minority	−0.481	0.344	0.142	0.517	0.370	0.105	1.250 **	1.978 **
	(−0.783)	(0.856)	(0.472)	(1.074)	(0.337)	(0.145)	(2.326)	(2.306)
Han–Han	0.593 **	0.957 ***	0.498 ***	0.551 *	1.013 *	1.328 ***	1.276 ***	0.595
	(2.015)	(4.057)	(2.704)	(1.849)	(1.927)	(3.153)	(3.867)	(1.119)
Han–Minority × Parents’ years of schooling	0.088				0.138			
	(1.229)				(1.078)			
Han–Han × Parents’ years of schooling	−0.009				0.047			
	(−0.229)				(0.651)			
Han–Minority × Non-agricultural work		−0.313				2.167 **		
		(−0.566)				(2.194)		
Han–Han× Non-agricultural work		−0.915 ***				−0.166		
		(−2.642)				(−0.269)		
Han–Minority × Non-rural hukou			1.145				0.828	
			(1.312)				(0.530)	
Han–Han × Non-rural hukou			0.637				0.110	
			(0.937)				(0.090)	
Han–Minority × Education expectation				−0.350				−0.906
				(−0.599)				(−0.872)
Han–Han × Education expectation				−0.018				0.941
				(−0.050)				(1.468)
Constant	3.776 ***	3.388 ***	3.775 ***	3.733 ***	2.119 **	1.986 **	1.938 **	2.494 ***
	(7.089)	(6.504)	(7.472)	(6.968)	(2.227)	(2.134)	(2.142)	(2.611)
Observations	1723	1723	1723	1723	1723	1723	1723	1723
R-squared	0.036	0.040	0.037	0.036	0.123	0.128	0.120	0.127
adj_R2	0.118	0.118	0.118	0.118	0.118	0.118	0.118	0.118

Note: *** *p* < 0.01, ** *p* < 0.05, and * *p* < 0.1.

## Data Availability

The data presented in this study are available upon request from the Insitute of Social Sceince Survey (ISSS) at http://www.isss.pku.edu.cn/cfps/en/index.htm accessed on 16 February 2023.

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
