# Peer review of "Parental Ethnicity and Adolescent Development: Evidence from a Nationally Representative Dataset"

_ijerph, 2023, doi:10.3390/ijerph20053799_

Round 1
Reviewer 1 Report
1. At line #223, the authors will define hukou.
2. In Statistical Analysis section (line #257-270), the authors’ application of multiple linear regression is inappropriate for modeling of panel data. The authors will need to use Stata Generalized Least Squares (GLS) random effect modeling procedures for panel data that takes into account of autocorrelations (i.e., correlations of measures across interview waves).
3. In the same section, the authors need preliminary analysis to test any potential multicollinearity among explanatory variables.
4. Since the inappropriate application of statistical procedures, the remaining sections (Results, Discussion, and Conclusion) will be modified according to new results from application of GLS.
5. The authors will report descriptive statistics for each wave of survey in Results section and Table 1.
6. In Discussion and Conclusion, the authors will provide policy implication for the findings on positive impact of Han-Han and Han-Minority family income on test scores and the implied difference of SES between Han and Minority families. Policy change may include improvement of Minority’s SES.
Reviewer 2 Report
I would like to thank the Editor for the opportunity to review this study and I am flattered to be able to provide my contribution. In general, I find this article to be well written, I do find this paper to be a good discussion issue about the relationship between parental ethnicity in the development of adolescents. However, the paper presents some minor weaknesses. I would ask the Authors to address minor amendments, as follows:
a) In the introduction (page 2, line 57) the authors should add the full stop.
b) Also, in the introduction (page 2), the authors should delete lines 75-77, as they repeat the usual structure of the articles and it is not necessary to explain the sections.
c) In the measures section, table 1, the authors should delete "mean" from the table 1, as they only include percentages in this table.
d) In the statistical analysis section (page 6, line 253), authors should delete the point above (See Figure 1).
e) In the results section (page 7), the authors should format the table better and capitalise the first letter of wordtest and mathtest.
Round 2
Reviewer 1 Report
The responses and revisions are appropriate and sufficient.